# Improving Forecast Accuracy with an Auto Machine Learning Post-Correction Technique in Northern Xinjiang

**Junjian Liu, Hailiang Zhang** 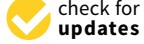**, Huoqing Li** and **Ali Mamtimin ***

Institute of Desert Meteorology, China Meteorological Administration, Urumqi 830002, China; liujj@idm.cn (J.L.); zhanghl@idm.cn (H.Z.); lihq@idm.cn (H.L.)
* Correspondence: ali@idm.cn

**Abstract:** Reliable meteorological forecasts of temperature and relative humidity are critically important to take necessary measures to avoid potential damage and losses. An operational meteorological forecast model based on the Weather Research and Forecast (WRF) model has been built in Xinjiang. Numerical forecasts usually have significant uncertainties and errors due to imperfections in models themselves. In this study, a straightforward automated machine learning (AutoML) approach has been developed to post-process the raw forecasts of the WRF model. The method was implemented and evaluated to post-process forecasts from 13 stations in northern Xinjiang. The post-processed temperature forecasts were significantly improved from the raw forecasts, with average RMSE values in the 13 stations decreasing from 3.24 °C to 2.34 °C by a large margin of 28%. As for relative humidity, the mean RMSE at 13 stations decreased from 19.54% to 11.54%, or it showed a percentage decrease of 41%. Meanwhile, biases were also significantly decreased, with average ME values being reduced from around 2 °C to ~0.33 °C for temperature and improved from −15.6% to ~0% for relative humidity. Moreover, forecast performance values after post-correction became much closer to each other than raw forecast performance values, improving forecast applicability at regional scales.

**Keywords:** post-correction; AutoML; WRF

## 1. Introduction

With continuous developments in numerical weather forecasting technologies, numerical weather models have become a major tool for operational weather forecasting [1]. Model forecasting accuracy is also directly related to weather forecasting accuracy and further affects the effectiveness of meteorological services [2,3]. At present, much research work is being carried out on numerical models such as the Weather Research and Forecast (WRF) model, including the optimization of physical parameterization schemes, the application of multi-source observation data, and improvements in assimilation algorithms [4]. However, because the initial and model boundary conditions, the physical process, the static surface data, and the mathematical description of the model framework all contain uncertainties, which bring errors into the numerical model, the output forecast is also subject to errors.

To further improve forecast accuracy, post-processing techniques have been widely used to reduce errors in operational forecasting applications [5]. These techniques include N-day running average correction [6], N-day Kalman filter post-processing techniques [7], and analog-based bias-correction methods [8]. These relatively straightforward methods often work with a small amount of data for bias correction. For example, bias-corrected models can gauge air temperature, relative humidity, and other surface meteorological variables from a WRF model with only eight previously selected forecasts [5]. The model output statistics (MOS) technique has also often been used to fuse previous model forecast errors with large amounts of data. For example, Kang et al. [9] improved air temperature forecast accuracy in a WRF model with an updated MOS method.

Recently, the development of machine learning methods has made it possible to digest even larger amounts of data with relatively high accuracy [10]. The most popular machine learning models include random forest [11], gradient boosting [12], and deep neural networks [13]. For example, Eccel et al. [14] used a multivariate machine learning model to post-process temperature forecasts. Machine learning models have also exhibited competitive capabilities in a wide range of prediction research works [15,16]. Studies have proven that the ensemble use of these models outperforms the use of each model alone [17]. Therefore, there is a need for ensemble models to improve model performance. However, this usually requires many trial-and-tests and various forms of expertise to tune up hyperparameters in order to determine how to construct an ensemble model from multiple candidate methods. To solve this tuning problem, in recent years, automatic an machine learning (AutoML) framework has been developing rapidly in the discipline of data science. Even though this approach is popular in model optimization [18], computer visions [19], and other areas, its application in MOS to improve model forecasting performance has been scarce.

In Xinjiang, a regional meteorological forecast model, which has run operationally at the Xinjiang Meteorological Bureau since 2018, was developed by the Institute of Desert Meteorology in the China Meteorological Administration in collaboration with the National Center for Atmospheric Research (NCAR), and it uses the Advanced Research WRF (ARW) research model and WRF data assimilation (WRFDA), version 3.8.1. Precipitation in Xinjiang has obvious terrain-dependent characteristics. The frequency of precipitation in the mountainous areas is highest where there are good forecast skills and fewer humans. Greater numbers of people from the oasis economic zone pay attention to the weather forecast in relation to temperature and relative humidity, both of which are important for travel and agricultural irrigation. Differences in topographic features between reality and the forecast model affect a forecast's precision. Forecast revision is still required.

This study used an AutoML-based ensemble model framework to post-process WRF temperature and relative humidity forecasts in the Urumqi-Changji-Shihezi (UCS) region, the core area of the oasis economic zone, in 2019.

## 2. Data and Method

### 2.1. Model and Configurations

The Advanced Research WRF (ARW) research model and WRF data assimilation (WRFDA), version 3.8.1 [20], were used for deterministic numerical weather forecasting. The fundamental configuration of WRF is listed in Table 1. There were two nested domains in the model, as shown in Figure 1. Domain 1 covered most of the Central Asia region, with a grid spacing of 9 km and $712 \times 532$ grid points. Domain 2 was centered on Xinjiang, China, with a grid spacing of 3 km and $832 \times 652$ grid points. There were 50 vertical computational layers with a pressure top of 10 hPa. Atmospheric and surface fields from the Global Forecast System of the National Centers for Environmental Prediction (GFS) model forecasts (the GFS forecasts) were introduced as the initial conditions. The physics schemes included WSM6 microphysics, ACM2 PBL, Kain-Fritsch deep convection, the RRTMG shortwave and longwave scheme, and the Unified Noah land-surface model. A three-dimensional variational (3DVar) technique was used for data assimilation. Conventional observations in 6 h cycling assimilation provided the analysis fields, including SYNOP (surface SYNOPtic observations, shown as the red points in Figure 1), METARs (METeorological Aerodrome Reports, airport ground-based), AMDAR (Aircraft Meteorological DAta Relay), and RAOB (RAwinsonde OBservations). In addition to conventional data, radial velocities from radar observations were also assimilated in Domain 2. Ground-level (2m) temperature and relative humidity were used as forecast variables in the post-processing procedures.

**Table 1.** Experimental configurations of WRF model.

| Model and Configurations | |
|---|---|
| **Model version** | ARWv3.8, nonhydrostatic=true |
| Domain 1 | 712 × 532, nominal 9 km |
| Domain 2 | 832 × 652, nominal 3 km |
| Vertical computation layers | 50 |
| Pressure top | 10 hPa |
| Lateral boundary conditions | GFS forecasts |
| Microphysics | WSM6 |
| Longwave radiation | RRTMG scheme |
| Shortwave radiation | RRTMG scheme |
| Land surface | Unified Noah land-surface model |
| Deep convection | Kain-Fritsch |
| Planetary-boundary and surface layer | ACM2 |

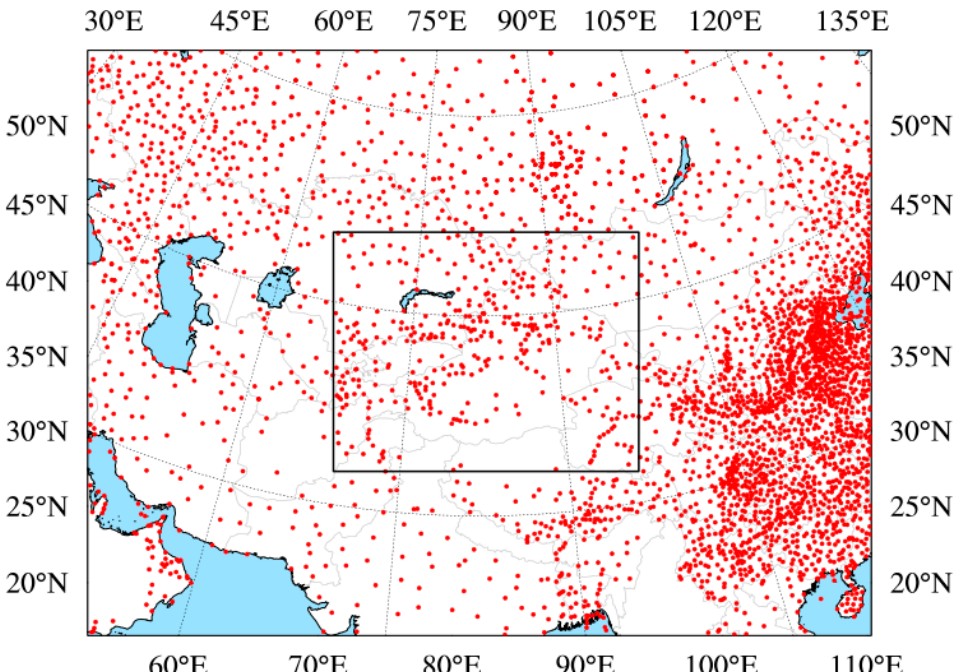

**Figure 1.** Forecast domains. The outside box indicates Domain 1 and the inner black box indicates Domain 2. The red dots refer to SYNOP stations.

### 2.2. Ground Meteorological Observations

Hourly mean observation data from 13 stations in the UCS area were used to evaluate the performance of the post-correction technique (Figure 2). The UCS area was selected because it is the core area of the oasis economic zone, which has large population and contains the Manas oasis agricultural area, the largest oasis agricultural area in Xinjiang. Meteorological conditions have a great influence on travel and agricultural irrigation. The meteorological variables included in this study were temperature and relative humidity. Hourly data were obtained from the China Integrated Meteorological Information Service System (CIMISS) of the China Meteorological Administration.

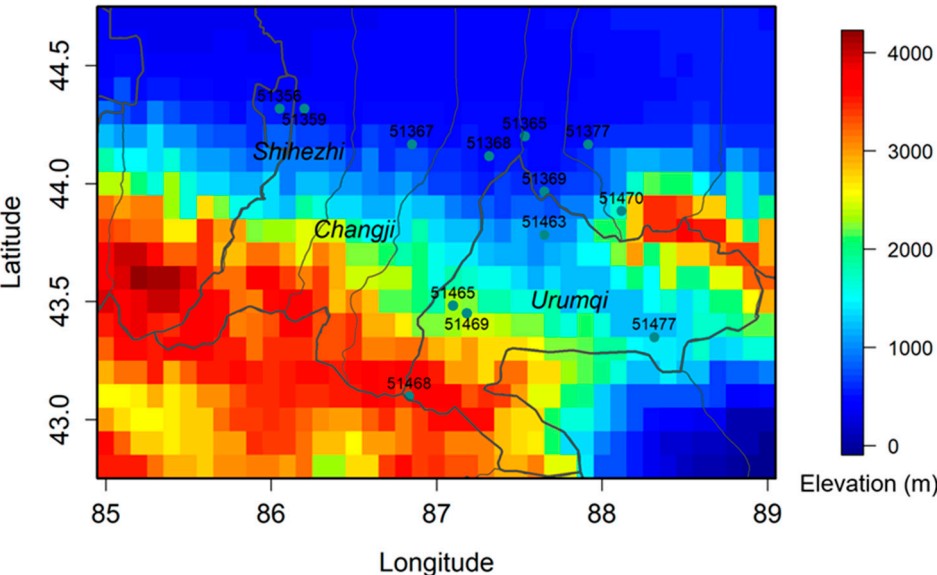

**Figure 2.** Stations in the UCS area used in this study. The bold polylines refer to city boundaries. The normal polylines refer to county boundaries.

### 2.3. AutoML Algorithm

The AutoML algorithm was developed from traditional ensemble models [21] by automatically performing model selection and optimizing parameterization. For example, in traditional ensemble models, several base models with fixed user-determined parameters were combined into a meta-model to improve the performance of each single model [22], as shown in the following equation:

$$Y = M_m(M_{b1}(X), M_{b2}(X), M_{b3}(X), \ldots, M_{bn}(X)), \tag{1}$$

where $Y$ and $X$ are, respectively, the response and explanatory data and $M_m$ and $M_{bi}$ ($i = 1, 2, \ldots, n$) are, respectively, the meta-model and the $i$-th base model. The base model outputs were fed into the meta-model, which automatically determined the weights of the base models according to predefined metrics (i.e., RMSE, NME). Generally, the base models with lower error levels had larger contribution weights in the final ensemble model. However, many experiments and trials are normally needed to determine a set of suitable base models and their parameter settings. For example, to use a random forest model (RF) as a base model, loss functions, the sampling approach, the number of trees, and other parameters must be determined. Parameter tuning is an even bigger problem for popular neural network models, which have many parameters that must be determined. Traditionally, their model parameters are generally defined and decided through manual tests or experienced expertise, which could be biased or limited in selecting the best potential models.

To solve this problem, an AutoML model was developed based on an ensemble learning framework by automatically optimizing the parameters of a group of pre-defined alternative base models [23]. These candidate base models include an RF model, gradient booster models (GBM), and a multiple layer perceptron model. The RF and GBM models are both composed of a group of relatively simple classifiers (trees) to construct competitive regression or classification capabilities, while they use different training strategies of bagging and boosting, respectively. In the AutoML model selection and training procedure, the automatic learning process achieved by the grid-search technique was implemented to optimize the hyperparameters of the base models [24,25]. The grid-search optimization technique used an exhaustive automatic "trial-and-error" method to select the best parameters from a pre-defined hyperparameter space. In other words, each hyperparameter within a model was tested with a group of candidate configurations. The hyperparameters

and corresponding model with best performances were determined as the final trained AutoML model.

To combine these base models, a stacking ensemble method was used to merge the selected base models to form an AutoML model. As shown in Figure 3, a previously fitted historical dataset was used to train the AutoML model before the model was used to post-process raw meteorological forecasts. Equation (2) shows how the temperature and relative humidity forecasts were post-processed:

$$Y_p = M_m\left(M_{b1}\left(RH_f,\ T_f\right), M_{b2}\left(RH_f,\ T_f\right), \ldots, M_{bn}\left(RH_f,\ T_f\right)\right), \tag{2}$$

where $M_m$ and $M_{bi}(i = 1,\ 2,\ 3\ldots,n)$ have same meaning as in Equation (1). The $RH_f$ and $T_f$ are input explanatory variables of the raw forecasts of relative humidity and temperature, respectively, and $Y_p$ is a post-processed meteorological variable, such as relative humidity or temperature. In the AutoML model used in this study, the number of base models was determined to be 15 after testing different numbers of base models to balance efficiency and accuracy.

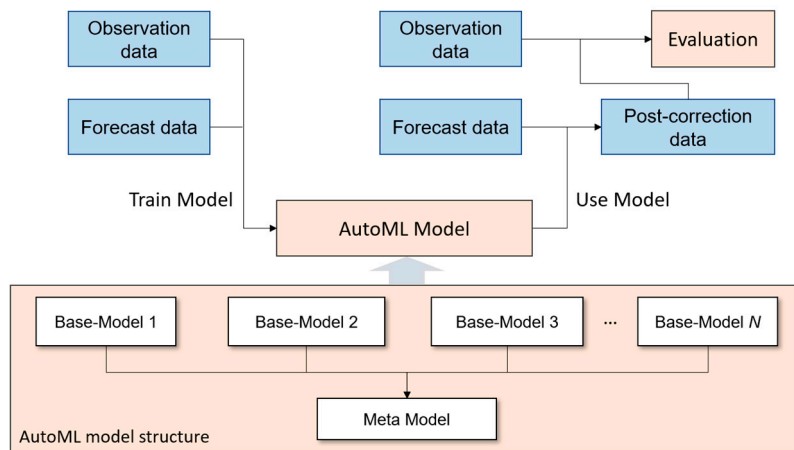

**Figure 3.** Stacking ensemble Auto-ML model framework.

### 2.4. Model Fitting and Evaluation

The hourly forecast data in August 2019 were used as independent assessment data to evaluate the performance of post-correction techniques. Two AutoML models were fitted, respectively, by the data from August in 2018, i.e., the same month to the evaluation period in the previous year and the data from July 2019, i.e., the previous month of the evaluation period; they were then compared.

For comparison, we used three other popular post-processing techniques for the bias-correction of forecasts in August 2019. First, a moving error post-processing method was used as the baseline model. Moreover, other MOS models of single linear regression (SLR), multiple linear regression (MLR), and random forest (RF) were also adopted for comparison. Furthermore, the confidence level that the AutoML model had a better performance than other models was tested and evaluated with the bootstrap method. In this method, around 250 data items were sampled from the evaluation data in August 2019 in 200 iterations. The sampling data in each iteration were evaluated with statistical metrics.

To evaluate post-correction performance, statistical metrics were reported, including coefficient of determination (R-square), mean error (ME), mean absolute error (MAE), normalized mean error (NME), and root mean square error (RMSE). Statistical metrics were

defined in Equations (3)–(7), where $y_i$ and $\hat{y}_i$ were, respectively, observation and forecast data items with N, and $\overline{y}$ was the average of N observations.

$$R^2 = 1 - \frac{\sum_i (y_i - \hat{y}_i)^2}{\sum_i (y_i - \overline{y})^2} \tag{3}$$

$$ME = \frac{1}{N} \sum_{i=1}^{N} (y_i - \hat{y}_i) \tag{4}$$

$$MAE = \frac{1}{N} \sum_{i=1}^{N} |y_i - \hat{y}_i| \tag{5}$$

$$NME = \frac{1}{N} \sum_{i=1}^{N} \frac{y_i - \hat{y}_i}{y_i} \tag{6}$$

$$RMSE = \sqrt{\frac{1}{N} \sum_{i=1}^{N} (y_i - \hat{y}_i)^2} \tag{7}$$

## 3. Results and Discussion

### 3.1. Raw Forecast Performance

The raw forecast performances at the 13 stations in the UCS area were similar. The data at station 51359 were used to analyze forecast performance as the correct data. As shown in Figure 4, the raw forecast performance for temperature and relative humidity varied temporally over the year. Generally, for temperature, the R-square values were lower in winter (December–January–February), with a mean of 0.65. In comparison, mean R-square values of temperature forecasts in summer was around 0.92. The larger RMSE values of around 5 °C indicated a more significant error in winter. The forecasts underestimated temperature during winter days, indicated by negative NME values of around −5 °C, and slightly overestimated temperature in summer days, with small positive values around 2 °C.

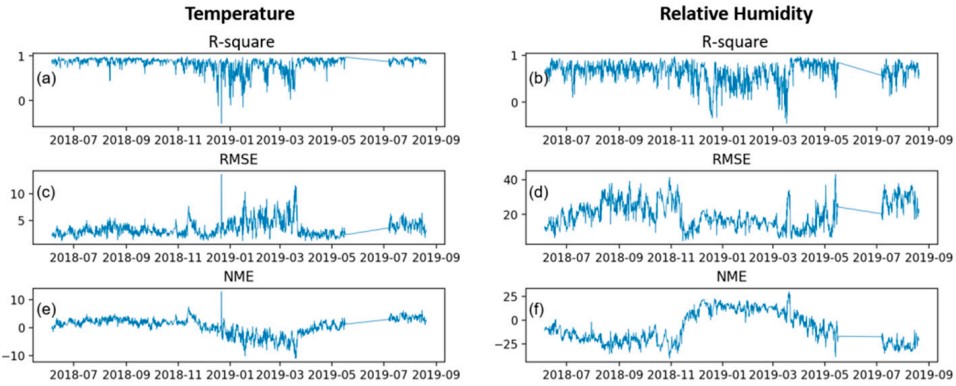

**Figure 4.** Forecast performance over time at station 51359: panels (**a**,**c**,**e**) are, respectively, referring to the R-square, RMSE, and NME values of temperature. The panels (**b**,**d**,**f**) are the metric values of relative humidity. The *x*-axis refers to observation time.

For relative humidity, the smaller R-square values were associated with weaker correlations between forecasts and observations in winter. However, error levels indicated by RMSE values were lower during winter days (around 15%), whereas the RMSE values in summer and autumn days were generally higher than 20%. As for NME, it was positive during winter days (around 10%), indicating an overestimation of relative humidity. During summer and autumn days, WRF underestimated relative humidity at this station, with NMEs being around −25%.

As shown in Figure 4, the values of forecast correlations and error levels changed rapidly during seasonal transitions. This indicated that using the time-moving approach to train the model for model forecast post-correction could have a good performance when error levels change rapidly day by day. Therefore, the aforementioned first model, as described in the last section, was expected to have a better performance in post-correction.

As shown in Figure 5, forecast metrics were evaluated at each forecast lead hour. For temperature, the R-square values decreased slightly from around 1 at the first lead hour to around 0.97 at the last lead hour. The R-square values linearly decreased in the first 40 lead hours before exhibiting periodic variations along the decreasing trend. The RMSE of the temperature forecast also increased from less than 3 °C to around 4.5 °C in the final lead hour. However, NME remained relatively stable for the first 40 lead hours before starting to fluctuate in a downward trajectory. As for relative humidity, it had a similar pattern to temperature, with a decreasing R-square, an increasing RMSE, and a fluctuating negative NME.

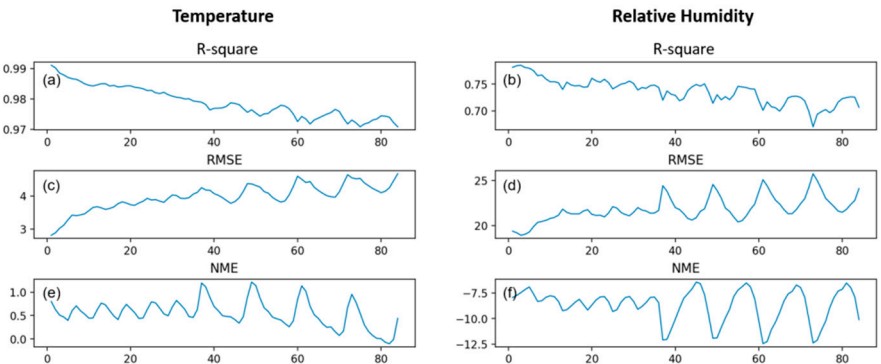

**Figure 5.** Forecast performance along the forecast lead hours at station 51359; panels (**a**,**c**,**e**) are, respectively, referring to the R-square, RMSE, and NME values of temperature. The panels (**b**,**d**,**f**) are the metric values of relative humidity. The *x*-axis refers to forecast lead time.

### 3.2. Post-Correction Performance

The base models, as automatically determined by the AutoML framework, are mostly GBM with different parameters, with a small proportion of other models such as general linear models. Within these models, GBM models usually have larger weights, which is similar to findings in previous studies [17].

Forecast performance after post-correction with AutoML generally improved at all 13 stations compared to raw forecast. The mean RMSE of post-corrected temperature forecast at the 13 stations was 2.34 °C, which was about 28% smaller than that of the raw forecast (3.24 °C) (Figure 6a). After correction, RMSE values among stations were all at similar levels, with a standard deviation of 0.34 °C, compared to 1.09 °C for raw forecasts. The largest improvement occurred at station 51470, where the percentage improvement was around 65% in terms of RMSE, from 6.02 °C to 2.11 °C. The smallest improvement happened at station 51469, where the RMSE increased by only 1%. As for the bias of the temperature forecasts, they were generally biased high, with an overestimation of 2.06 °C for all stations. The largest bias occurred at station 51470, where the temperature was overestimated by 5.68 °C. The smallest bias occurred at station 51469, with an MAE value of 0.70 °C. After post-correction by the AutoML model, mean MAE values became much smaller, with a mean of 0.33 °C, and the overestimation at most stations was less than 1 °C.

As for the relative humidity forecast, the mean RMSE of the raw forecast was 19.54% for all 13 stations, with mean ME values being around −16.21%. After post-correction, mean RMSE became smaller (11.54%), which was a 41% decrease. The greatest improvement was at station 51359, where the RMSE improved from 29.31% to 12.53%, i.e., a decrease of 57%. Similar performance was also achieved at stations 51368 and 51356, with RMSE decreasing by 56% and 53%, respectively. As for MAE values, raw WRF forecasts greatly

overestimated relative humidity at all 13 stations, with a mean ME of −15.41%. The most significant underestimation of the raw forecast occurred at station 51359, which had an ME of −26.3%. After correction, forecasts were generally unbiased, with the largest bias (−3.52%) occurring at station 51365.

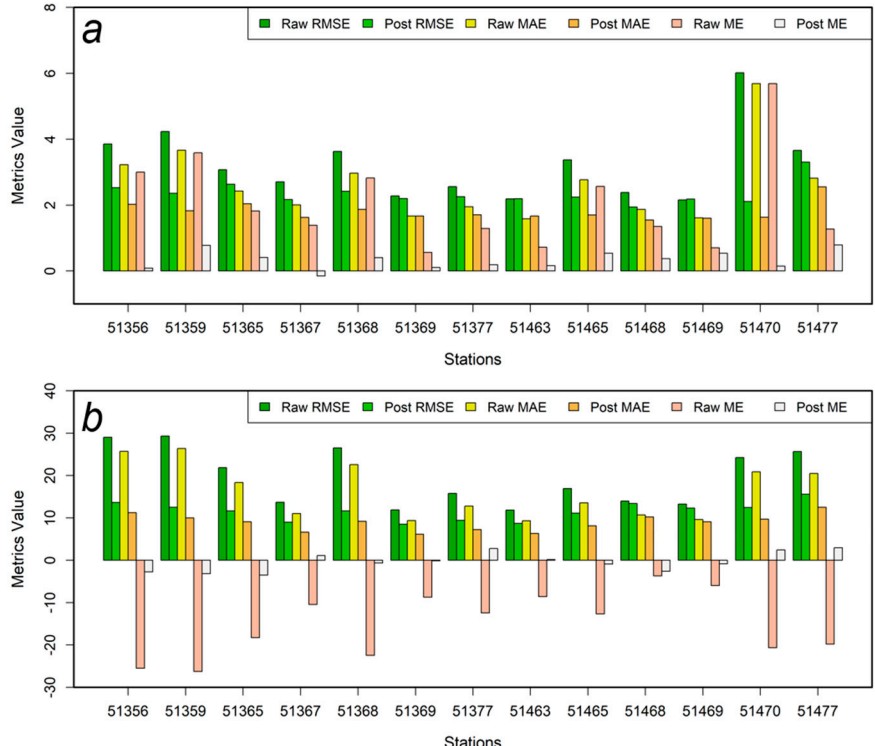

**Figure 6.** Forecast performance of raw forecast and post-correction results for: (**a**) temperature and (**b**) relative humidity.

It should be noted that, for both temperature and relative humidity, forecast biases were much smaller in almost all stations. This revealed that the AutoML post-correction model could substantially remove systematic errors in raw forecasts. At the stations where bias values of raw forecasts were small, such as 51463 for temperature and 51468 for relative humidity, the post-correction performance evaluated by RMSE improved less than other stations. This indicated the post-correction models are relatively weak when it comes to reducing random errors in raw forecasts.

Compared with the post-correction performance of the models fitted by data from the previous month, the model fitted by data from August 2018 generally behaved better, as shown in Figure 7. As for temperature, the post-correction performance of the latter model also had a larger RMSE than the model of the previous month at most stations.

Moreover, compared to three other MOS methods, our ensembled AutoML model also had a better performance, as shown in Figures 8–10. For example, RF, MLR, and moving error post-correction model, respectively, have larger RMSE values than the AutoML model, with a mean difference of 0.11, 0.07, and 0.09 °C for temperature and 0.55%, 0.61%, and 1.12% for relative humidity. Besides, compared to the SLR model, the AutoML model also showed a better performance than the MLR model, as exhibited in Figure 11. The post-correction improvement due to AutoML compared to SLR was −0.17 °C and −1.54% for T and RH RMSE values, respectively; this was more significant than the improvements of of 0.04 °C and −0.56% caused by MLR.

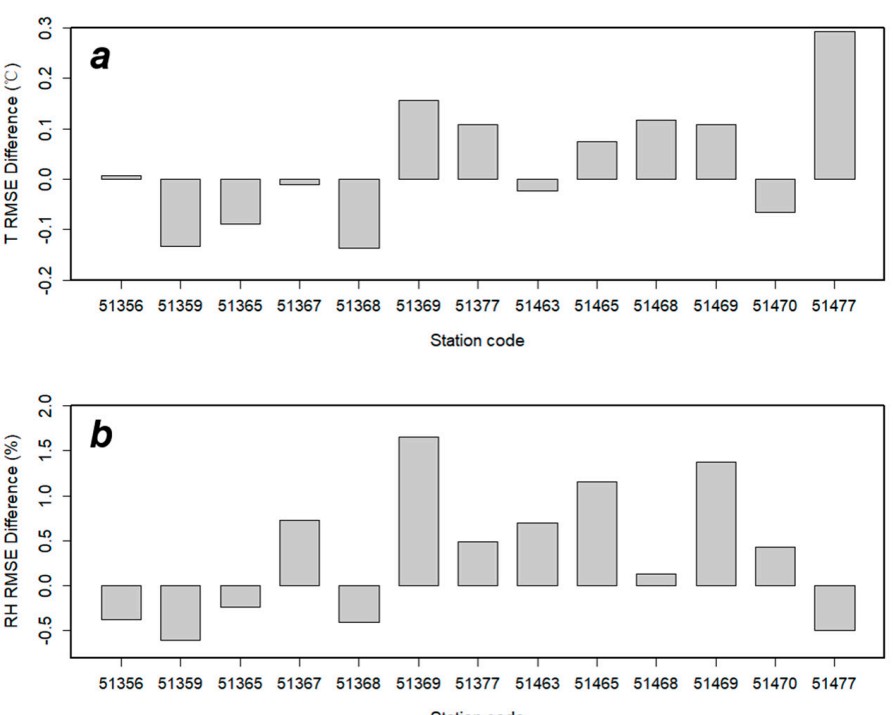

**Figure 7.** Difference between the RMSE values of post-correction for (**a**) temperature and (**b**) relative humidity by AutoML model fitted by previous month in the same year (2019) and historical data in last year (2018).

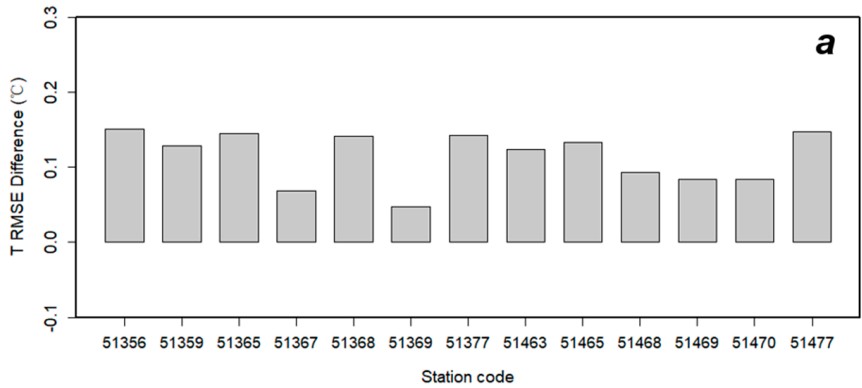

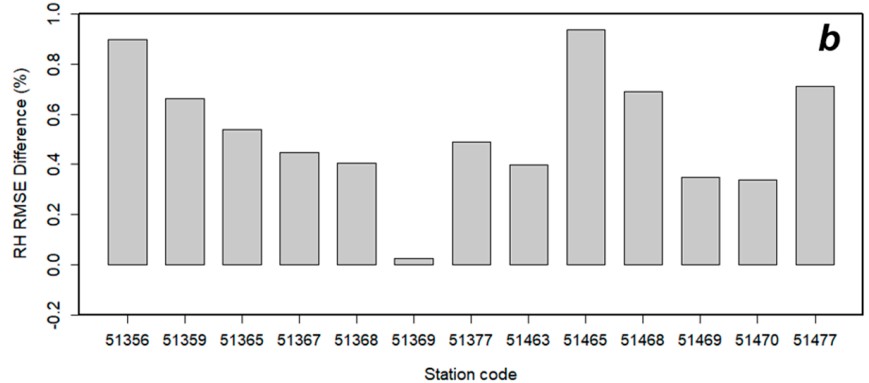

**Figure 8.** Difference between the RMSE values of post-correction forecasts for: (**a**) temperature and (**b**) relative humidity by AutoML model and random forest model.

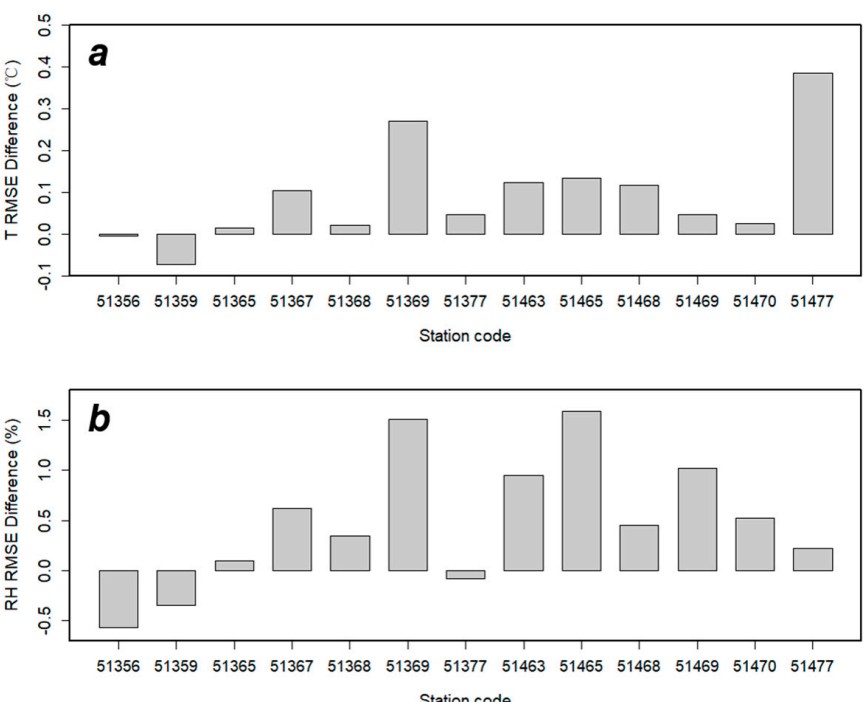

**Figure 9.** Difference between the RMSE values of post-correction forecasts for: (**a**) temperature and (**b**) relative humidity by AutoML model and traditional linear MOS method.

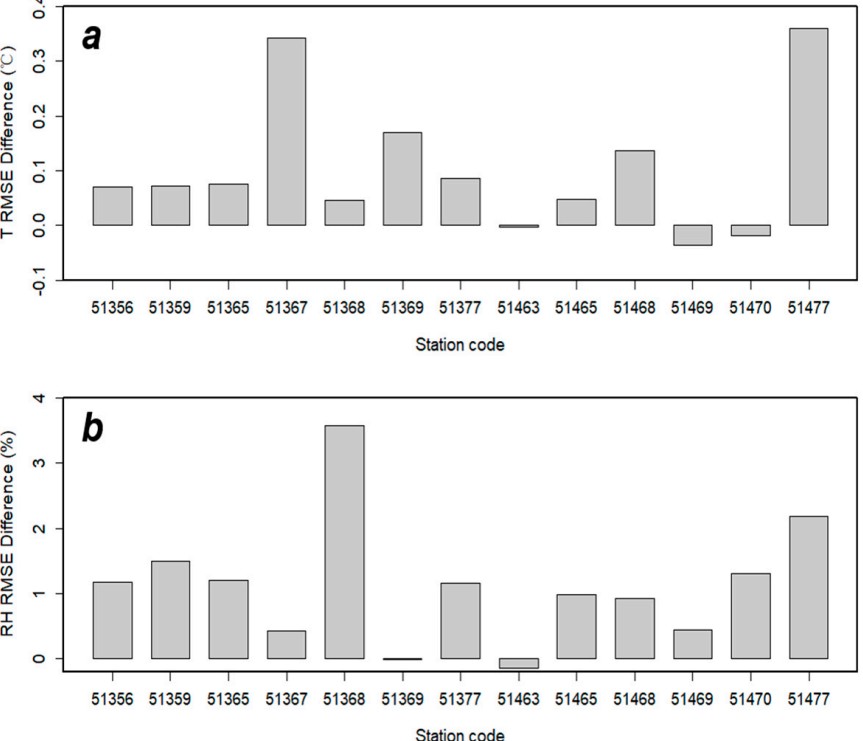

**Figure 10.** Difference between the RMSE values of post-correction forecasts for: (**a**) temperature and (**b**) relative humidity by AutoML model and moving error post-correction method.

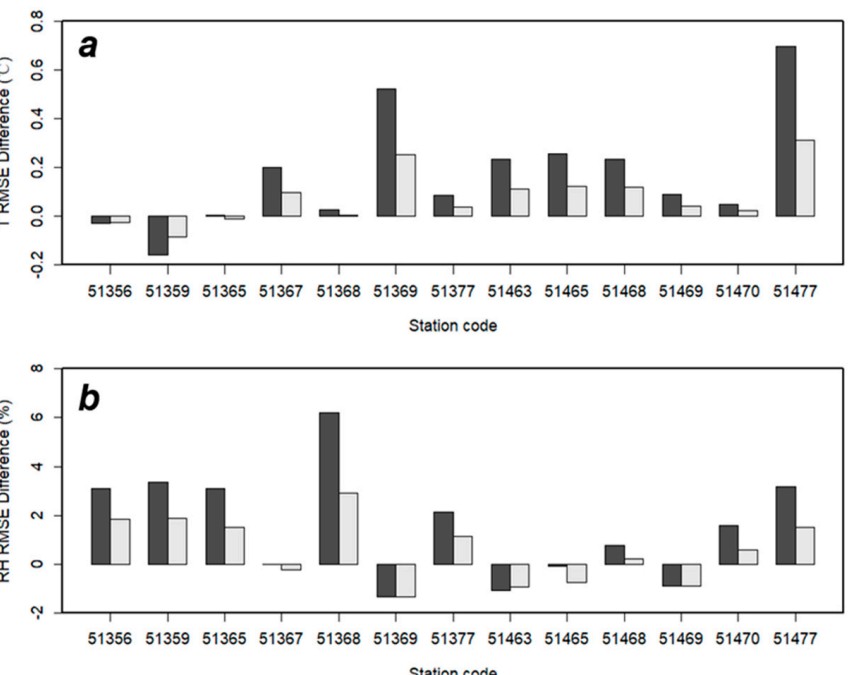

**Figure 11.** Difference between the RMSE values of post-correction forecasts for: (**a**) temperature and (**b**) relative humidity by AutoML model and multiple linear regression MOS method compared to single linear regression method. The deep gray bar and light gray bar indicate raw forecasts and post-processed forecasts.

Results evaluated with the bootstrap method also indicated a better performance of the AutoML in most stations, as indicated by Figures 12 and 13. However, it should be noted that post-correction performance was evaluated with one month of data, which could have limitations in terms of representing long-term performance, especially when in relation to climate events such as the El Niño-Southern Oscillation.

*3.3. Time-Series Analysis*

The weather forecast model generates forecast results routinely four times per day at UTC 0:00, 6:00, 12:00, and 18:00. Post-correction performance was significantly improved from the original forecasts in almost all forecast processes, as shown in Figure 14 for station 51359. At this station, the RMSE of the raw temperature forecast was around 4 °C, with a mean of 4.01 °C and a standard deviation of 1.2 °C. After post-correction, RMSE decreased significantly, with a mean of 2.1 °C and a standard deviation of 0.57 °C. For relative humidity, the RMSE of the raw forecast was relatively large, with a mean of 26% and a standard deviation of 8%. This deviation indicated that raw forecasts have low stability in relation to model performance, which will decrease users' confidence when using their data in applications. Specifically, RMSE decreased to much lower levels, with a mean of 13% and a standard deviation of 3%. The decreased mean RMSE indicated that the forecast accuracies had improved for temperature and relative humidity. Meanwhile, the reduced standard deviation of RMSE indicated an enhanced stability of forecast performance along different forecast processes. Even though post-correction forecasts showed a deteriorating trend along leading hours, a complete circle of forecasts was corrected all at once rather than by each lead hour, as shown in Figure 15. Our approach will also improve public users' confidence in forecasts because the errors are stable.

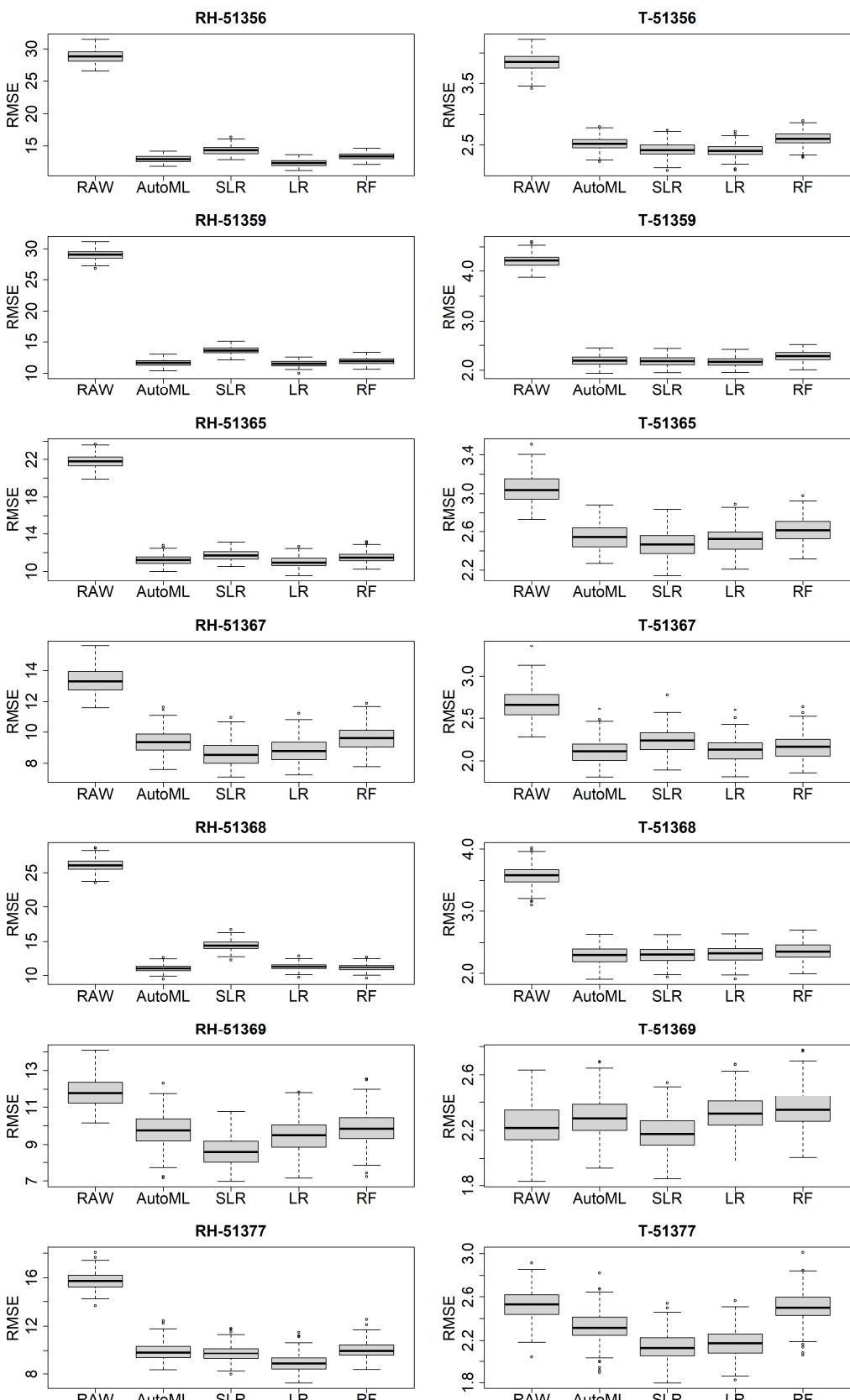

**Figure 12.** *Cont.*

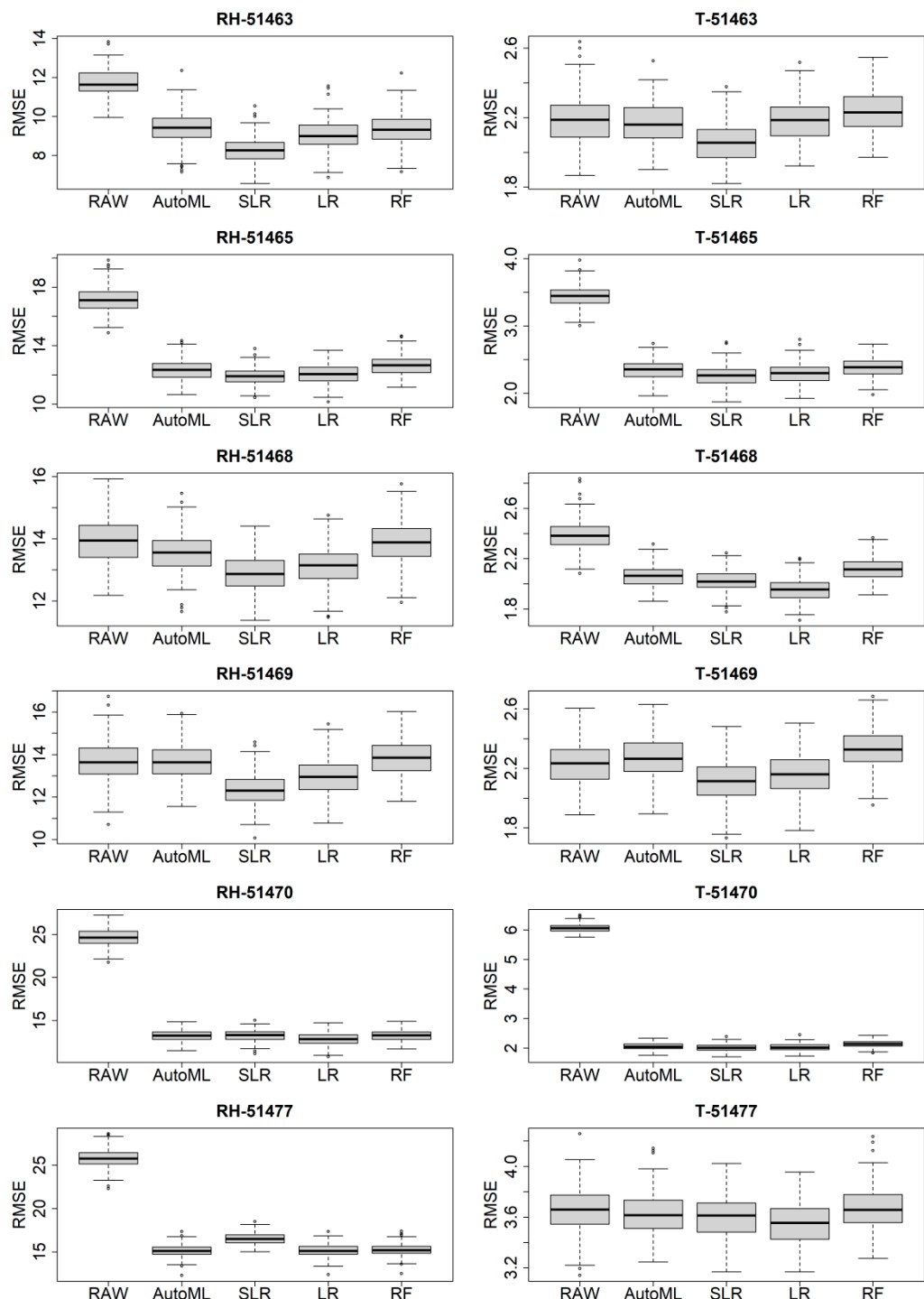

**Figure 12.** The boxplot of RMSE values by raw forecast, AutoML, SLR, LR, and RF models evaluated with bootstrap sampling method.

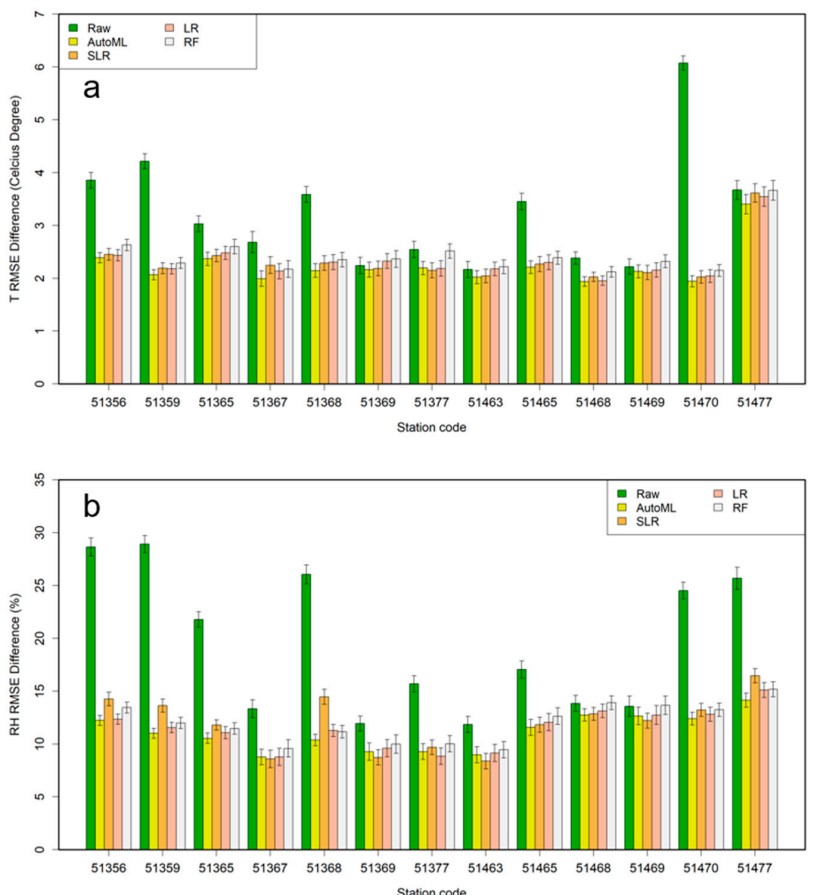

**Figure 13.** The bar-plot of mean RMSE values by raw forecast for (**a**) temperature and (**b**) relative humidity, AutoML, SLR, LR, and RF models evaluated with bootstrap sampling method. The uncertainty bar was added.

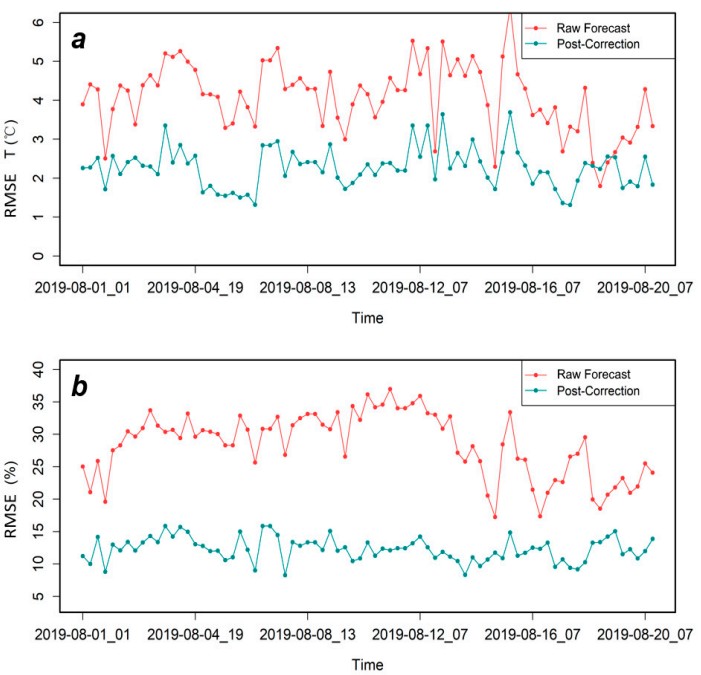

**Figure 14.** RMSE of raw forecast and post-correction results in each forecast period for: (**a**) temperature and (**b**) relative humidity at station 51359.

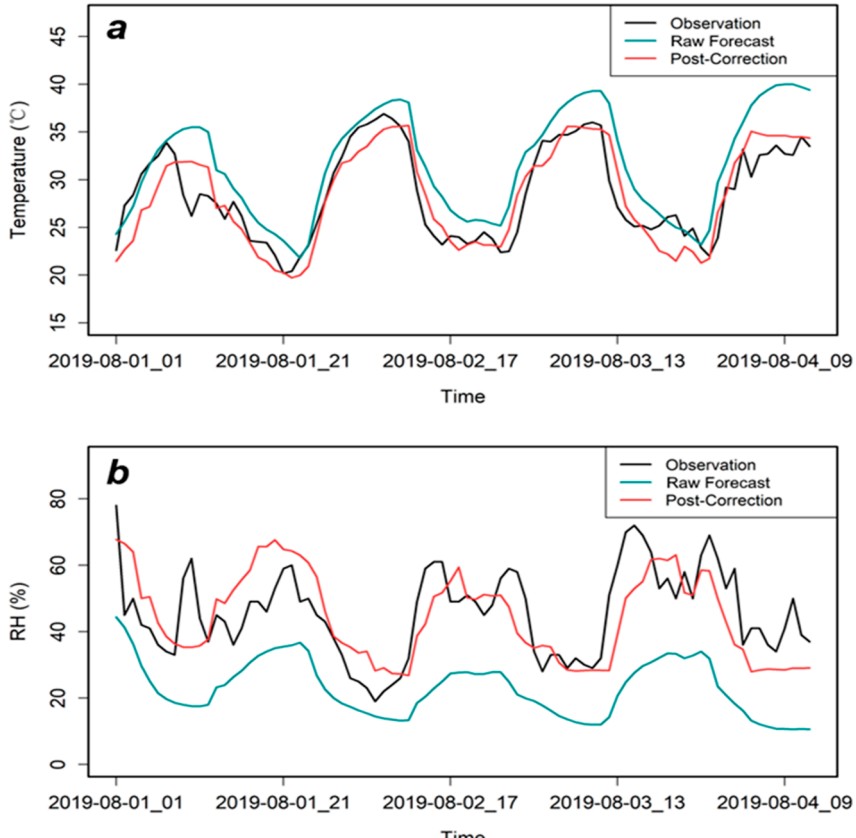

**Figure 15.** Comparison of the raw forecast with post-corrections to observations for: (**a**) temperature and (**b**) RH of a forecast process at station 51359 starting at 0:00 on 1 August 2019.

## 4. Conclusions

Meteorology forecasts from the raw WRF model have relatively high error levels. Previous studies have demonstrated that forecast performance could be improved by post-correction processes. In this aspect, this study built and tested a straightforward and auto-optimizing AutoML model to bias-correct the raw forecasts. The AutoML is not only strong in terms of modeling capabilities, but it is also easy to construct and fit due to its automatic workflow. This method was implemented and evaluated to post-process forecasts from 13 stations in northern Xinjiang. In general, forecast performances were significantly improved over raw forecasts. For temperature, the average RMSE values decreased from 3.24 °C to 2.34 °C, which was a large percentage decrease of 28%. As for relative humidity, mean RMSE values at the 13 stations decreased from 19.54% to 11.54%, or a percentage change of 41%. In addition, biases were also significantly decreased. The post-processed forecasts also have a relatively enhanced performance stability, which will help in forecast data applications.

Compared to three other MOS methods, the ensembled AutoML model also had a better performance with lower error levels. RF, MLR, and moving error post-correction models, respectively, have larger RMSE values than the AutoML model, with mean differences of 0.11, 0.07, and 0.09 °C for temperature and 0.55%, 0.61%, and 1.12% for relative humidity. The post-correction performance was significantly improved from the original forecasts in almost all forecast processes. A complete circle of forecasts was corrected all at once rather than by each lead hour.

The model could be readily used in post-processing forecast field data by training it with multiple stations within a region. It should also be applicable to other forecast variables. In terms of limitations, the model is generally time consuming compared to a

single model, since it needs to test many parameters. The post-correction model could be updated on a weekly or monthly time scale.

**Author Contributions:** Conceptualization A.M.; methodology, J.L.; software, J.L. and H.L.; validation, H.Z. and H.L.; formal analysis, H.L.; investigation, A.M.; writing, original draft preparation, J.L.; writing, review and editing, J.L. All authors have read and agreed to the published version of the manuscript.

**Funding:** The National Key Research and Development Program of China, grant number 2018YFC1507105; the National Natural Science Foundation of China, grant number 41875023, 41801019; Central Scientific Research Institute of the Public Basic Scientific Research Business Professional, grant number IDM2018010.

**Data Availability Statement:** The data used in this paper can be provided upon request by reaching Junjian Liu (liujj@idm.cn).

**Acknowledgments:** Thanks to Baolei Lyu, a respectable, responsible and resourceful scholar, who has provided me with valuable guidance in every stage of the writing of this paper.

**Conflicts of Interest:** The authors declare no conflict of interest.

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
