# Peer review of "Improving Forecast Accuracy with an Auto Machine Learning Post-Correction Technique in Northern Xinjiang"

_applsci, doi:10.3390/app11177931_

Round 1

Reviewer 1 Report

There are some weaknesses through the manuscript which need improvement. Therefore, the submitted manuscript cannot be accepted for publication in this form, but it has a chance of acceptance after a minor revision. My comments and suggestions are as follows:

1- Abstract gives information on the main feature of the performed study, but a couple of sentences about the background of the study must be added.

2- Authors must clarify necessity of the performed research. Objectives of the study must be clearly mentioned in introduction.

3- The literature study must be enriched. In this respect, authors must read and refer to the following papers: (a) machine learning in mechanical engineering: https://doi.org/10.1016/j.jmrt.2021.07.004  (b)  machine learning in oil and gas: https://doi.org/10.1016/j.ptlrs.2021.05.009 The current version of introduction is too short.

4- Authors must clarify why this area was selected to test the developed system.

5- The main reference of each formula must be cited. Moreover, each parameters in equations must be introduced. Please double check this issue. The main formula of RMSE should be presented.

6- Why the specified time (e.g., August 2019, July 2019) was considered for evaluation.

7- In several images (e.g., Fig. 12), the text and legend are illegible. The lager and high quality figures are required.

8- Reason for deviation is the results (presented curves) must be discussed.

9- In its language layer, the manuscript should be considered for English language editing. There are sentences which have to be rewritten.

10- The conclusion must be more than just a summary of the manuscript. List of references must be updated based on the proposed papers. Please provide all changes by red color in the revised version.

Reviewer 2 Report

The manuscript assessed the performance of AutoML approach to post-process the meteorological forecasts data from WFR model. The proposed model showed good performance to improve the quality of temperature and relative humidity. In my opinion, the manuscript can be considered acceptable for publication in this journal after major revisions. Please see an attachment. 

Reviewer 3 Report

I have read the paper titled; “Improving Forecast Accuracy with an Auto Machine Learning Post-Correction Technique in Northern Xinjiang”. It is well written but there is a need to carefully read through it again by the authors correcting all the typos. 1. What do the red dots in Fig 1 represent? If these are the SYNOP, kindly mention it here in the figure’s caption 2. What does the x-axis in Fig. 4 & 5 represent? The lead hour time? 3. Correct legend for Figure 11 4. Try to read the paper and correct all typo errors. 5. How is Figure 1 related to China in General 6. Line 85-86, the authors said: “The UCS area was selected because the local population is large and the meteorological conditions have great influence” This is weak, therefore what is the usefulness of this study or what is the motivation? Applications in heat waves, extreme events, agriculture, hydrlogic modeling etc? 7. The introduction section looks weak. Could the authors include other studies done in China or within the region that have used this technique? Or is this first method/study applying this in China 8. In the Results and discussion section as well, similar works done in other regions of the world are not included. 9. In the conclusion, would similar results be obtained using other stations? 10. How many stations were used for training and testing the model?

Round 2

Reviewer 2 Report

Thank you for your effort in revising the manuscript. I have only one minor comments that please add all equations of evaluation metrics (R-square, ME, MAE, NME). You have only one equation (RMSE) in your paper. 

Thanks,

Author Response

Thanks.The equations of evaluation metrics (R-square, ME, MAE, NME) were  add

in the 2.4. Model Fitting and Evaluation in Lines 178~180.
